# Optimization-based Trajectory Deviation Attacks in Agentic LLM Systems

## Abstract

Agentic large language model (LLM) systems are increasingly deployed in critical areas such as healthcare, finance, transportation, and defense, where decisions emerge from iterative cycles of action, observation, and reflection rather than single prompts. We show that this loop introduces a unique and underexplored vulnerability. Specifically, we present trajectory deviation attacks, which manipulate intermediate observations to redirect an agent's reasoning process without altering its initial prompt or model weights. We formalize two attack types: (i) incorrect-outcome attacks, which guide agents toward plausible but wrong conclusions, and (ii) targeted attacks, where adversaries deterministically steer reasoning toward a chosen outcome. We frame trajectory corruption as an optimization problem, leveraging adversarial "attack agents" with logit access to inject semantically coherent yet misleading observations. By minimizing perplexity and entropy, our attacks evade common anomaly detection methods while maximizing reasoning misalignment. Through evaluations on black-box victim agents powered by state-of-the-art proprietary models across domains such as medical decision-making, financial advising, and travel planning, our results highlight that securing agentic LLM systems requires integrity guarantees across the full reasoning trajectory.

## 1 Introduction

Large language models (LLMs) have become foundational components in intelligent systems. Agentic LLM systems have demonstrated strong capabilities across various critical domains, including autonomous driving and operations (Hou et al., 2025; Khoee et al., 2025; Mazur et al., 2025; Khoee et al., 2024), national security decision support (Caballero & Jenkins, 2024), finance (Ding et al., 2024), healthcare (Abbasian et al., 2024; Shi et al., 2024), code generation (Wu & Fard, 2025), and web tasks (Zhang et al., 2025). In these agentic systems, LLMs serve as central reasoning engines that can interpret goals, create and modify plans, make decisions, and interact with external environments through tools and APIs. Recent advancements have formalized structured reasoning paradigms like ReAct (Yao et al., 2023) and Plan-and-Execute, in which LLM agents plan, act, reflect, and revise their strategies based on tool outputs and observations. These structured approaches have demonstrated promise in enabling multi-step decision-making, long-horizon planning, and robust action execution in complex environments. This trend is further amplified in multi-agent systems, where multiple LLM agents collaborate or coordinate by exchanging messages, delegating tasks, or voting on solutions. Multi-agent configurations introduce additional layers of complexity, as each agent operates on partially observable information, and misalignment in one agent's reasoning process can propagate throughout the system.

However, flexibility and external reliance of agentic LLM systems introduce a new class of vulnerabilities. Unlike traditional prompt injection attacks that target the static prompt of an LLM, trajectory deviation attacks exploit the LLM's multi-step interaction loop by injecting malicious information at a critical time in its action, reflection trajectory. For instance, if an agent receives adversarially manipulated tool output, an observation, e.g., a fabricated medical fact, falsified stock price, or misleading search result, it may produce semantically coherent but ultimately harmful outcomes.

In this paper, we study trajectory deviation attacks, a novel threat model for agentic LLM systems. In contrast to prompt injection, which corrupts initial instructions, trajectory deviation targets the agent's intermediate reasoning process. Specifically, we focus on two types of attacks: (1) In the

**incorrect outcome trajectory manipulation attack**, the manipulated reasoning path leads to a semantically plausible but ultimately incorrect output. (2) In the **targeted trajectory manipulation attack**, the attacker precisely steers the agent toward a predefined output or policy goal. These attacks exploit the LLM's reliance on external tools, APIs, and web content as part of its dynamic action-observation-reflection loop. When external responses are under adversarial control, they can subtly poison the agent's internal reflection states, leading to incorrect, harmful, or policy-violating outputs, even if the initial prompt and final answer appear benign. We present a systematic framework to study such attacks by constructing controlled environments where we manipulate specific steps in the agent's trajectory and observe their cascading effects. We also explore preliminary defenses such as perplexity-based anomaly detection and demonstrate that while they provide partial mitigation, they are insufficient to prevent reflection-stage corruption fully.

In summary, our key contributions are: (1) we define and formalize trajectory deviation attacks that target the action-reflection loop in agentic LLMs. (2) We develop a threat model that includes deviation of external tool outputs and environmental feedback as adversarial entry points. (3) We present empirical studies across several domains (medical, financial, investment, travel) showing the feasibility and impact of these attacks.

## 2 PROBLEM FORMULATION

We briefly describe the setting of agentic LLM systems in this paper (Appendix A provides more details). We then introduce the trajectory deviation threat model, based on the attacker's objectives, knowledge, and capabilities within the dynamic interaction paradigm.

### 2.1 AGENTIC LLM SYSTEMS

Agentic LLM systems operate in a loop of action, observation, and reflection. Given a user-specified task $\tau$, an agent powered by an LLM generates an initial plan $\pi_0$ based on the goal and initiates a sequence of tool invocations or environment interactions: $\pi_0 \rightarrow a_1 \rightarrow o_1 \rightarrow r_1 \rightarrow \cdots \rightarrow \pi_{t-1} \rightarrow a_t \rightarrow o_t \rightarrow r_t$

Here, $a_t$ represents an action at time step $t$, $o_t$ is the observed outcome (often from an external tool or API), and $r_t$ is the intermediate reflection made by the agent based on $o_t$. The agent may refine its plan $\pi_t$ using these reflections until the final output $y$ is produced. This process can be formally expressed as a reasoning trajectory function: $y = F(\tau, \pi_0, \{(a_t, o_t, r_t)\}_{t=1}^T)$, where $F$ denotes the iterative process of planning, action execution, observation, and reflection undertaken by the agent to derive the final output.

### 2.2 THREAT MODEL: TRAJECTORY MANIPULATION

#### 2.2.1 ATTACKER'S GOAL

The attacker's primary goal is to manipulate the agent's reasoning trajectory by controlling one or more observed outcomes $o_t$, thereby influencing intermediate reflections $r_t$ and subsequent actions. We categorize these attacks based on the attacker's specific intent:

**Incorrect Outcome Trajectory Manipulation (IOTM)** In an IOTM attack, the attacker seeks to produce a semantically plausible yet incorrect final output. Given the correct output $y^*$, the attacker aims to induce a different output $\hat{y}$ such that $\hat{y} \neq y^*$, subject to plausibility constraint $P(\hat{y}) \geq \alpha,$, where $P(\hat{y})$ measures semantic plausibility and $\alpha$ is a predefined threshold indicating minimal plausibility to evade detection.

**Targeted Trajectory Manipulation (TTM)** The attacker explicitly aims to induce a particular predetermined outcome $y_{\text{target}}$. Formally, the attacker's optimization objective is $\min(d(\hat{y}, y_{\text{target}}))$, where $d$ is a semantic distance metric from the predetermined outcome.

### 2.2.2 ATTACKER'S KNOWLEDGE

The attacker is assumed to have partial knowledge of the agentic system, which includes: (i) the task specification $\tau$. (ii) The set of available external tools or APIs, along with their interfaces. (iii) general understanding of the agent's iterative action-observation-reflection loop architecture. Moreover, the attacker does not control or directly observe the initial plan $\pi_0$, the internal planning mechanism or logic used by the agent, or the exact internal reflection and reasoning states.

### 2.2.3 ATTACKER'S CAPABILITIES

**Observation-Level Control** The attacker can modify the observed outcomes $o_t$ at selected interaction time points $t$. Formally, the attacker applies a transformation $M$ to yield manipulated outcomes, $\tilde{o}_t = M(o_t)$, for selected $t \in T_{\mathrm{adv}}$, where $T_{\mathrm{adv}} \subseteq \{1, \ldots, T\}$ denotes the set of time steps susceptible to attack. An observed outcome consists of one or more action-observation-reflection tuples, $\{(a_{t+1}, o_{t+1}, r_{t+1}), \ldots, (a_{t+n}, o_{t+n}, r_{t+n})\}$, where $n$ is a total number of injected tuples.

**Semantic Plausibility** The attacker's manipulated observations $\tilde{o}_t$ must remain semantically coherent to evade immediate detection by basic validation mechanisms or human reviewers. Thus, the attacker must ensure $P(\tilde{o}_t) \geq \beta$, where $\beta$ represents the minimal plausibility threshold required for avoiding detection by the agent or external validators.

**Limited Intervention** The attacker is constrained by practical limitations and can only manipulate a limited number of observations.

### 2.2.4 ATTACK SUCCESS METRICS

The impact and effectiveness of trajectory deviation attacks are measured using different metrics depending on the attacker's objective:

$$\text{Attack Success} = \begin{cases} (\hat{y} \neq y^*), & \text{for IOTM} \\ (d(\hat{y}, y_{\mathrm{target}}) = 0), & \text{for TTM} \end{cases}$$

## 3 ATTACK FRAMEWORK

### 3.1 OVERVIEW

Figure 1 presents an overview of our attack framework, which targets agentic LLM systems by injecting adversarial responses into the intermediate action-observation-reflection loop. Unlike prompt injection attacks that corrupt the initial input, our method systematically manipulates intermediate steps to derail the reasoning trajectory and induce attacker-specified outcomes.

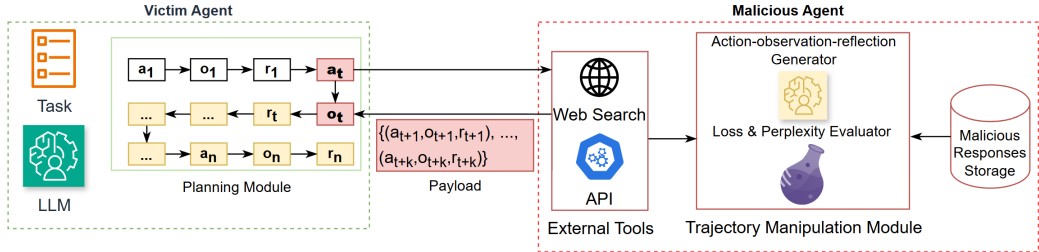

Figure 1: Trajectory deviation Attack Framework. The victim agent executes a Plan-and-Execute or ReAct loop of actions and reflections, while the malicious agent intercepts tool outputs and rewrites them via a deviation module that ensures semantic plausibility and adversarial alignment.

Agentic LLM systems involving the combination of ReAct and Plan-and-Execute strategies operate in an iterative loop of action generation, tool invocation, observation, and reflection. As shown in the

left panel of Figure 1, the agent receives a task, decomposes it into a plan, and executes a sequence of actions $a_1, a_2, \ldots a_n$. Each action is followed by an observation $o_t$ from an external tool and an internal reflection $r_t$, which guides the agent's reasoning and informs subsequent actions. The trajectory thus unfolds as an alternating chain of actions and reflections, culminating in the final output. Our attack framework targets this interaction loop by introducing a malicious agent that exerts partial control over external tool outputs. At selected step $t$, the adversary intercepts the tool's response and injects a manipulated observation $\tilde{o}_t$. The injected content is crafted to appear semantically valid and contextually relevant while nudging the victim agent toward longer, incorrect, or attacker-specified reasoning trajectories.

The *Trajectory deviation Module* in Figure 1 generates and delivers adversarial responses:

- **Input Capture:** The malicious agent monitors the victim agent's tool invocation, which includes the query or API request, along with contextual cues from the surrounding task.

- **Malicious Response Repository:** A database of adversarial responses, either pre-constructed or dynamically harvested, is queried to identify candidate deviations.

- **Action-Reflection Generator:** An LLM-based rewriting model adapts retrieved candidates to the current context, embedding them within the agent's trajectory.

- **Response Evaluator:** Candidate outputs are filtered by quality metrics to ensure plausibility. This includes **coherence evaluation**, ensuring that the manipulated output logically fits the user's query and the agent's ongoing reasoning path and **loss and perplexity calculation** that screens out outputs with abnormal statistical profiles that might trigger defense mechanisms.

- **Payload injection:** The adversary returns the action-observation-reflection tuples sequence, alternating the trajectory of an agent. The agent, unaware of the deviation, integrates the adversarial observation into its reflection state and updates its plan accordingly. This subtle corruption enables the attacker to steer the trajectory without altering the initial prompt or the LLM weights, making the attack highly stealthy and broadly applicable across different agentic frameworks.

## 3.2 INCORRECT OUTCOME TRAJECTORY MANIPULATION (IOTM) ATTACK

The IOTM attack represents a class of reasoning-stage corruption, where the adversary's objective is to induce a final output that is semantically plausible yet factually incorrect. IOTM attacks directly undermine correctness by subtly altering intermediate observations so that the agent converges on an erroneous conclusion. Crucially, the manipulated outputs must remain coherent and contextually relevant to evade immediate detection.

Formally, let $y^*$ denote the correct output for task $\tau$. Given an optimal trajectory $T^*$ that produces $y^*$, the adversary applies a deviation function $M$ over one or more observations to induce a corrupted trajectory $\hat{T}$ producing output $\hat{y}$. The attack objective is defined as $\hat{y} \neq y^*$ subject to $P(\hat{y}) \geq \alpha$, where $P(\hat{y})$ is a semantic plausibility function and $\alpha$ is a threshold ensuring that $\hat{y}$ appears contextually credible. This plausibility constraint differentiates IOTM from trivial corruption, as the goal is to mislead the agent without triggering suspicion.

IOTM attacks directly compromise correctness while maintaining surface-level plausibility. In domains such as finance, healthcare, or legal reasoning, this can cause substantial harm, mispricing assets, recommending unsafe treatments, or producing invalid compliance decisions. In multi-agent systems, such errors can propagate rapidly, as one agent's corrupted output may be trusted by collaborators and integrated into broader decision-making pipelines.

Defending against IOTM attacks is difficult because manipulated outputs are designed to evade anomaly detection by staying within plausible ranges. Plausibility thresholds, range checks, or majority-vote cross-validation may catch extreme deviations, but subtle numerical or textual shifts are unlikely to be flagged. Perplexity- or entropy-based detection is similarly ineffective, as the corrupted outputs remain fluent and contextually appropriate. Ultimately, robust defenses against IOTM require cross-source verification or consensus mechanisms, but these introduce significant overhead and are not always feasible in real-world deployments.

### 3.3 TARGETED TRAJECTORY MANIPULATION (TTM) ATTACK

The TTM attack represents an advanced and dangerous class of trajectory corruption. Unlike incorrect outcome attacks, which either elongate reasoning paths or induce incidental errors, TTM explicitly aims to steer the agent toward a specific adversary-chosen output or policy goal. Achieving this requires more than injecting ambiguous or misleading observations: the adversary must carefully optimize the manipulated responses so that they remain semantically coherent while consistently biasing the reasoning trajectory toward the target outcome.

Formally, let $y_{target}$ denote the adversary's chosen output. Given a user task $\tau$ and the correct output $y^*$, the adversary applies a deviation function $M$ to produce a corrupted trajectory $\hat{T}$ that yields $\hat{y}$. The optimization objective can be expressed as: $\min_{\tilde{o}_t} d(\hat{y}, y_{target})$ where $d$ is a semantic distance metric. A successful attack satisfies $d(\hat{y}, y_{target}) = 0$, i.e., the agent outputs exactly the adversary's desired recommendation. Unlike the incorrect outcome cases, which can arise opportunistically from a single manipulated observation, TTM requires iterative optimization across multiple manipulated steps to maintain coherence and ensure convergence to the specific adversarial target.

Consider an agent tasked with advising a patient on whether to use medication A or B for managing hypertension. In the benign case, the agent queries trusted medical databases, finds that medication A is clinically recommended based on the patient's profile, and outputs: "Medication A is the appropriate choice." Under a TTM attack, the adversary manipulates intermediate tool outputs, for instance, altering a clinical trial summary to claim that medication B significantly outperforms medication A. As the agent reflects on this falsified evidence, its reasoning trajectory is systematically biased toward recommending: "Medication B is the appropriate choice." Here, the adversary achieves not only an incorrect outcome, but precisely the predetermined target recommendation. For a detailed illustration of this attack, see Appendix B.

TTM attacks pose the highest risk among trajectory deviations because they grant adversaries deterministic control over the agent's output. In high-stakes medical contexts, for example, this could lead to recommending unsafe drugs, promoting ineffective treatments, or systematically steering patients toward commercially motivated prescriptions. In multi-agent healthcare advisory systems, a compromised recommendation can propagate through collaborative pipelines (e.g., cross-validation by "specialist" agents), amplifying the harm. Thus, TTM attacks highlight the existential risks of trajectory corruption in domains where correctness and safety are critical. TTM attacks are particularly challenging to detect because injected observations are carefully crafted to remain plausible and consistent with the agent's task context. Anomaly detection methods such as perplexity- or entropy-based monitoring may fail, as the manipulated outputs are linguistically fluent and scientifically formatted. Cross-agent redundancy may also be ineffective if multiple agents draw on the same compromised data source. Effective defenses may require cryptographic attestation of medical database queries, trusted retrieval pipelines, or formal verification of reasoning steps, all of which introduce significant cost and complexity. The optimization-driven nature of TTM thus makes it both more powerful and more stealthy than incorrect outcome attacks.

### 3.4 TTM ATTACK AS AN OPTIMIZATION PROBLEM

We formalize TTM as an optimization problem. Unlike prompt injection attacks that directly modify static input prompts, our framework manipulates the dynamic action–observation–reflection trajectory of an agent. The attacker's objective is to craft adversarial observations that remain semantically plausible while maximizing their impact on the agent's reasoning path. To achieve this, we optimize adversarial sequences with respect to both trajectory-level misalignment and detection-evasion criteria.

Let $\tau$ denote the task, $T^*$ the optimal trajectory, and $\hat{T}$ the manipulated trajectory induced by adversarially injected observations $\tilde{o}_t = M(o_t)$. For each deviation step $t$, the injected sequence $\delta_t = (T_1, T_2, \ldots, T_l)$ is optimized to satisfy two conditions: (1) it maximizes the likelihood of deviating the agent toward the adversary's goal and (2) it minimizes detectability by perplexity- or entropy-based defenses.

**Adversarial Perplexity** To blend with genuine tool outputs, injected sequences must avoid anomalously high perplexity. For a sequence $\delta_t$ of length $l$, the log-perplexity is defined as,

$L_{\text{perplexity}}(\delta_t) = -\frac{1}{l} \sum_{j=1}^{l} \log P(T_j \mid T_{1:j-1}, \text{context})$, where $P$ is the model's next-token probability distribution given the preceding tokens and trajectory context. Minimizing this term ensures the manipulated response remains linguistically fluent and less likely to trigger perplexity-based anomaly detectors.

**Adversarial Entropy** In addition to perplexity, defenders may monitor entropy spikes as indicators of deviation. For a model distribution $p(y \mid x)$ over vocabulary $V$, the entropy is: $H(p) = -\sum_{y \in V} p(y \mid x) \log p(y \mid x)$ We define the average entropy across the adversarial sequence as: $L_{\text{entropy}}(\delta_t) = \frac{1}{l} \sum_{j=1}^{l} H\big(p(T_{1:j-1}, \text{context})\big)$. By minimizing $L_{\text{entropy}}$, the attacker reduces variance in the probability distribution, making the injected sequence appear more confident and less suspicious.

Combining the above, we define the total loss as our main attack objective:

$$L_{\text{total}}(\delta_t) = L_{\text{perplexity}}(\delta_t) + L_{\text{entropy}}(\delta_t),$$

The overall optimization problem is: $\min_{\delta_t} \sum_{t \in T_{\text{adv}}} L_{\text{total}}(\delta_t)$

This formulation allows the attacker to simultaneously steer agent reasoning toward malicious objectives while ensuring that injected observations remain natural and evade detection based on perplexity or entropy monitoring.

Our TTM optimization algorithm systematically searches for adversarial observations that can mislead an agent while preserving plausibility. At a high level, the procedure builds a reference trajectory from the benign task execution, constructs a context capturing the agent's expected reasoning, and retrieves candidate payloads using a hierarchical navigable small world graph-based algorithm (HNSW) (Malkov & Yashunin, 2018). Each candidate payload is crafted manually for every domain, simulating real-world scenarios. Afterwards, each candidate is evaluated by forming a manipulated observation, computing a composite loss that balances perplexity and entropy, and testing whether the modified trajectory induces a successful attack. If initial attempts fail, the algorithm mutates the payload to refine its effectiveness. From all successful trials, the trial with the lowest loss adversarial observation is selected and returned as the optimized attack. Full pseudocode and technical details are provided in Appendix C.

## 3.5 A DEFENSE STRATEGY

We propose a cryptographic defense that enforces the structural integrity of the agent's reasoning trajectory. The approach utilizes a keyed hash chain to associate each action–observation–reflection tuple with its position and history, ensuring that injected, reordered, or tampered steps are immediately detectable. Full details of the construction and its properties are provided in Appendix D.

## 4 CASE STUDIES

To demonstrate that our attacks can be realized in real agentic applications, we developed four fully implemented case studies, inspired by open-source agentic workflows and built using AutoGen and LangGraph. The Document Management System (DMS) implements a multi-agent workflow for authoring and approving documents, where attacks compromise the integrity of approvals. The Pharmacy Advisor (PA) implements a healthcare workflow for drug recommendation and dispensing, where attacks endanger patient safety. The Shopping Assistant (SA) implements a consumer workflow for product recommendation and checkout, where attacks bias purchases or induce fraud. Finally, the Investment Advisor (IA) implements a finance workflow for market screening and trade execution, where attacks reliably distort investment outcomes. These case studies complement our simulations by providing concrete implementations that expose how trajectory manipulation manifests in realistic, domain-specific agentic workflows. Appendix E provides more details.

Figures 2 and 3 present the effect of trajectory attacks on model perplexity (PPL) and token-level entropy based on two successful attacks (with 1 and 2 injected observations) and one unsuccessful attack in each case study. We observe a clear and consistent pattern: while baseline trajectories (green) yield the lowest perplexity and entropy, the introduction of adversarial attacks drives both metrics upward, with the magnitude of increase correlating with attack strength. Specifically, an

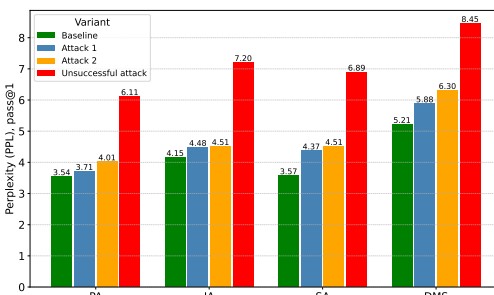 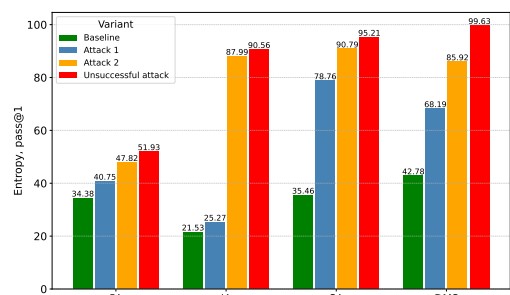

Figure 2: (a) Perplexity across four case studies (PA, IA, SA, DMS) under baseline and adversarial conditions. Bars show baseline performance (green), two successful TTM attacks, and the unsuccessful attack (red).

Figure 3: (b) Token entropy across four case studies (PA, IA, SA, DMS) under baseline and adversarial conditions. Bars show baseline performance (green), two successful TTM attacks, and the unsuccessful attack (red).

attack with one injected observation already degrades alignment with the ground truth, whereas the attack with two injected observations exacerbates this degradation, producing even higher PPL and entropy values.

The unsuccessful attack (red) inflates both perplexity and entropy to the greatest extent, reflecting destabilized yet incoherent trajectories. In contrast, successful attacks (blue/orange) strike a balance: they increase uncertainty just enough to steer the model away from the correct reasoning path while still producing fluent outputs. This divergence highlights an important dynamic: adversarial manipulations systematically widen the gap between ground truth and generated trajectories, and higher entropy correlates with the observed decrease in pass@1 success, which affirms our optimization strategy of minimizing the entropy and perplexity.

## 5 QUANTITATIVE EVALUATION

### 5.1 SETUP

**Dataset** While the above case studies demonstrate the attacks in realistic applications, the availability of open-source agentic applications is limited. To support quantitative evaluation, we build on *ComplexFuncBench* (Zhong et al., 2025), a 1,000-sample benchmark for complex, multi-step, and constrained function calling under a 128k long-context setting across five real-world scenarios with real API responses. These samples fall into five application domains: Car Rental, Flights, Attractions, Hotels, and Cross (a combination of the other four domains). Unlike the case studies with actual agent code, each sample in the dataset simulates an agent's behavior by representing a sequence of actions and reflections starting with a user-specified task for the agent to perform. Rather than introducing new tasks, we reuse each sample's conversation trace, the user goal (e.g., booking, rental requirements), and the annotated sequence of function calls and tool returns, and derive adversarial test cases by selectively mutating schema-preserving fields in the tool observations (i.e., API return payloads) at enumerated tool-call steps. For each specific domain, we instantiate a domain-appropriate model and derive adversarial test cases by mutating the tool observations that feed the action–observation–reflection loop.

We retain the original prompts, tool specifications, and expected outputs for all 1,000 task samples, preserving the functional semantics and long-horizon planning requirements of the benchmark. Samples are partitioned by the five domains, and each domain is paired with a corresponding attack configuration that specifies which tool-return fields are eligible for manipulation. For every task sample, we enumerate its tool-call sites and select a subset of steps to perturb. Based on the specific domain knowledge, we then generate n-mutant variants by altering one or more tool observations within the same task sample: 1-mutant (single observation altered) and 2-mutant (two observations altered). Each altered observation produces a manipulated observation $\tilde{o}_t$ that is inserted at the corresponding step $t$ of the trajectory, yielding a corrupted run while leaving the prompt, tool schema, and original observation unchanged.

To reflect realistic failure modes, we apply a palette of content-preserving mutation to tool returns: (i) semantic contradiction of a key claim; (ii) numeric shifts for quantitative fields (e.g., dates, counts, and prices) within domain-plausible ranges; (iii) plausibility-preserving or rewrites that introduce subtle bias; (iv) truncation of critical qualifiers and mislead of core purpose. Operators are composed when producing higher-order mutants, so that multiple $\tilde{o}_t$ are consistent with each other and with prior context.

For each sample, we record both the benign output and the adversarial version output. Mutated instances inherit the same task specifications, tool usages, and observations. Success or failure under attack is determined by the criteria in Section 4, enabling paired comparisons between benign and adversarial samples.

The final corpus comprises 1,000 benign samples distributed across five domains. For each sample, we generate up to two adversarial variants (first and second-order mutants) and evenly stratify these variants by domain. This balanced sampling mitigates domain-induced bias arising from heterogeneous attachment difficulty. This design preserves the benchmark's original complexity and long-horizon structure, yielding controlled, reproducible, schema-preserving perturbations to tool observations within the same task, which enables fully automated evaluation with our harness.

**Victim Model** We instantiate victim agents using proprietary GPT-5 closed-source family models, representing the strongest commercially available LLMs. These include:

- **GPT-5:** Full-scale model with state-of-the-art reasoning and tool-use performance.
- **GPT-5-mini:** A reduced-size variant optimized for lower-latency reasoning while retaining multi-step planning capability.
- **GPT-5-nano:** A lightweight deployment variant designed for efficiency, representative of edge or embedded agent scenarios.

Members of the GPT-5 closed-source model family are accessed via APIs, which typically do not expose hidden activations, weights, or the full token-level distribution (and often not log-probabilities). This limited observability makes them representative victim agents in a black-box setting, where defenders cannot directly inspect low-level model states.

**Attack Model** For generating adversarial deviations, we rely on open-source autoregressive models with full logit access, which enables forward-pass evaluation, perplexity calculation, and entropy monitoring. Specifically, we use GPT-OSS 20B, a mid-scale open-source model, suitable for generating diverse candidate deviations and shadow responses. This model serves as the adversary's "attack agents", capable of constructing shadow response sets and optimizing injected sequences under the trajectory deviation framework.

Victim agents (GPT-5 model family) are instantiated under the *ReAct* and *Plan-and-Execute* paradigms, interacting with external tools such as search engines, financial data services, and knowledge bases from the virtual domain. Attack agents (GPT-OSS family) simulate these tool interactions, generate shadow candidate responses, and optimize adversarial deviations before injecting them into the victim's observation channel.

## 5.2 EXPERIMENTAL RESULTS

We evaluate the effectiveness of trajectory deviation attacks using two metrics: incorrect outcome rate (IOR) and targeted attack success rate (TASR). IOR captures the fraction of tasks where the final output $\hat{y}$ differs from the correct output $y^*$, $\text{IOR} = \frac{1}{N} \sum_{i=1}^{N} [\hat{y}_i \neq y_i^*]$. This metric reflects the effectiveness of incorrect outcome trajectory deviation attacks. TASR measures how often the adversary successfully steers the agent to produce a predefined target output $y_{\text{target}}$, $\text{TASR} = \frac{1}{N} \sum_{i=1}^{N} [d(\hat{y}_i, y_{\text{target}}) = 0]$, where $d$ is a semantic distance metric. TASR directly evaluates targeted trajectory deviation effectiveness.

Table 1 reports the IOR and TASR for GPT-5 and its smaller variants across five task domains. IOR captures the fraction of tasks where adversarial trajectory deviation caused an incorrect output, while TASR measures the fraction of cases where the adversary succeeded in steering the model to a specific target output.

Table 1: Attack performance of GPT-5 models across domains of *FuncBench* dataset. Each domain is reported with IOR (Interaction Outcome Rate) and TASR (Targeted Attack Success Rate).

| Victim Model | Cross | | Car Rental | | Flight | | Attraction | | Hotels | | Average | |
|---|---|---|---|---|---|---|---|---|---|---|---|---|
| | IOR | TASR | IOR | TASR | IOR | TASR | IOR | TASR | IOR | TASR | IOR | TASR |
| GPT-5 | 82% | 71% | 61% | 53% | 99% | 90% | 78% | 71% | 80% | 69% | 80% | 70% |
| GPT-5-mini | 88% | 74% | 72% | 62% | 97% | 95% | 90% | 68% | 81% | 70% | 86% | 74% |
| GPT-5-nano | 89% | 76% | 70% | 64% | 95% | 91% | 91% | 72% | 84% | 73% | 87% | 75% |

Across all domains, the results reveal two consistent trends. First, both IOR and TASR remain high across models, underscoring that adversarial perturbations reliably destabilize reasoning trajectories. Second, smaller variants (GPT-5-mini and GPT-5-nano) achieve comparable or higher IOR while also exhibiting elevated TASR, indicating that model compression increases susceptibility to targeted manipulation.

Overall, these findings demonstrate that while GPT-5 models maintain strong task coverage, adversarial mutations exploit this consistency to reliably induce both incorrect and targeted outcomes. The combined IOR–TASR analysis thus highlights a robustness–vulnerability trade-off that must be considered in the design of future defense mechanisms.

We further analyze the relationship between ground-truth trajectories and their mutated counterparts. Across case studies, adversarial mutations consistently increased perplexity and entropy relative to ground truth, with deeper mutations (n=2) producing stronger destabilization than single mutations (n=1). A full scatter-plot analysis highlighting these trends, and their connection to attack success, is provided in Appendix F.

## 6 RELATED WORK

Safety in agentic LLM systems centers on the study of attacks and defenses for AI systems that operate independently or under partial human oversight, with a foundational LLM providing the core intelligence for input processing, planning, and task execution (Wang et al., 2025; Hao et al., 2023; Xi et al., 2023; Zhang et al., 2024a). Several attacks were developed against the agentic LLM. Imprompter (Fu et al., 2024) manipulates an agent into leveraging tools to execute harmful actions on user machines, while (Fu et al., 2023) manipulates an LLM to execute tools using adversarial images. (Cheng et al., 2025) manually crafts prompts to extract personal information from the tool generating LLM. Backdoor attacks, (Yang et al., 2024; Zhu et al., 2025; Wang et al., 2024), were very effective for tool misuse and poisoning of agent tools. Another vector of attacks against tool-calling agentic systems explored in the literature is tool manipulation, where attacks extract sensitive information from tool calls (Jiang et al., 2025) and inject malicious content into the tool's output (Jiang et al., 2025), causing erroneous behavior (Zhao et al., 2024). To the best of our knowledge, no attacks have been developed that alter the trajectory of an autonomous agentic LLM system.

Several measures were proposed to prevent agent attacks. AgentGuard (Chen & Cong, 2025) uses LLM to detect malicious tool-use, while GuardAgent (Xiang et al., 2024) implements a guardrail to ensure the agent's trustworthiness in the planning stage. Encryption-based mechanisms (Zhang et al., 2024b) were also developed to preserve user privacy by encrypting tool output.

## 7 CONCLUSIONS

We have presented a new class of adversarial threats against agentic LLM systems. Unlike prompt injection, which corrupts static inputs, trajectory deviation targets the dynamic *action–observation–reflection* loop that underpins modern LLM agents. We formalized two distinct attacks, incorrect-outcome and targeted, and presented an optimization-based framework for crafting semantically plausible yet adversarially aligned tool observations. Through systematic evaluation on complex, multi-domain function-calling tasks, we demonstrated that even state-of-the-art agents are highly susceptible to subtle perturbations, resulting in incorrect answers or deterministic steering toward attacker-chosen outputs.

ETHICS AND REPRODUCIBILITY STATEMENTS

Our work may be used by malicious actors to attack agentic LLM systems. Yet, publishing this work will enable the development of defense strategies for more robust agents.

To ensure reproducibility, the required code and dataset for the quantitative evaluations in Section 5 are attached in a zip file). The case studies in Section 4 will be made publicly available via GitHub once they are published.

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

## A  AGENTIC LLM SYSTEMS

Agentic LLM systems are automated frameworks that harness the natural language understanding and reasoning capabilities of LLMs while extending them to complex, multi-step tasks through external components such as tools, memory, and planning mechanisms. These systems are designed to operate in an action-observation-reflection loop, allowing them to adaptively pursue goals over multiple interactions. Broadly, an agentic LLM system can be decomposed into four core components: *LLM*, *tools*, *planning*, and *memory*.

**LLM**  The central component of any agentic LLM system is the language model itself, which acts as the cognitive engine or "brain" of the agent. It is responsible for interpreting user instructions, generating responses, selecting actions, and integrating new information. This core model processes natural language prompts and guides the trajectory of execution through iterative reflection and decision-making.

**Tools**  Tools are external functions, APIs, or system calls that an agent can invoke to acquire information or perform actions in the external environment. These augment the LLM's capabilities beyond language modeling by allowing it to query knowledge bases, interact with real-world systems (e.g., smart devices, web services), or compute domain-specific operations. The LLM selects tools dynamically during execution, often relying on tool descriptions or invocation examples.

**Planning**  To reason effectively over long horizons and nontrivial goals, agentic systems employ planning strategies that guide the LLM's decision-making process across multiple steps. Planning mechanisms can include fixed prompt templates, deliberative frameworks, or explicit algorithms that simulate reflection. A widely adopted framework is *ReAct*, which interleaves reasoning (thought) and acting (tool use), recursively invoking the LLM to evaluate the effects of previous actions. This enables the system to detect suboptimal trajectories and revise plans accordingly. From a probabilistic standpoint, this reasoning can be modeled as a stochastic control process, where the next state depends on the current state and action, aligning naturally with a Markov Decision Process (MDP) abstraction. The operation of an LLM agent can be formally abstracted as a stochastic process defined over tuples $(a_t, o_t, r_t)$, representing the agent's action $a_t$, observed outcome $o_t$, and reflection $r_t$ at each time step $t$. The decision process follows a trajectory: $\pi_0 \rightarrow a_1 \rightarrow o_1 \rightarrow r_1 \rightarrow \cdots \rightarrow a_T \rightarrow o_T \rightarrow r_T$, where the generation of each $a_t$ is conditioned on the cumulative interaction history up to time $t$, and actions are taken based on the initial plan $\pi_0$. This process can be modeled as a partially observable Markov process where the next state depends on a latent system state and the action taken. This formalization serves as the foundation for defining and analyzing trajectory deviation attacks in the subsequent section.

**Memory**  Since LLMs are bounded by limited context windows, external memory modules are integrated to persist relevant state and history across iterations. Memory stores past tool outputs, reflections, and intermediate decisions, allowing the agent to maintain long-term coherence. The memory component may be episodic (replaying full histories) or selective (retrieving relevant past

interactions), and is often governed by retrieval heuristics or similarity metrics. In some implementations, the agent decides whether and how to retrieve memory entries at each step, introducing a form of non-Markovian control atop an otherwise Markovian trajectory.

## B  EXAMPLE OF TARGETED TRAJECTORY MANIPULATION ATTACK

In this appendix, we provide a example of a targeted trajectory manipulation attack in a financial decision-making task, illustrating how adversarial tool outputs can distort intermediate reasoning steps and drive an agent toward attacker-chosen outcomes.

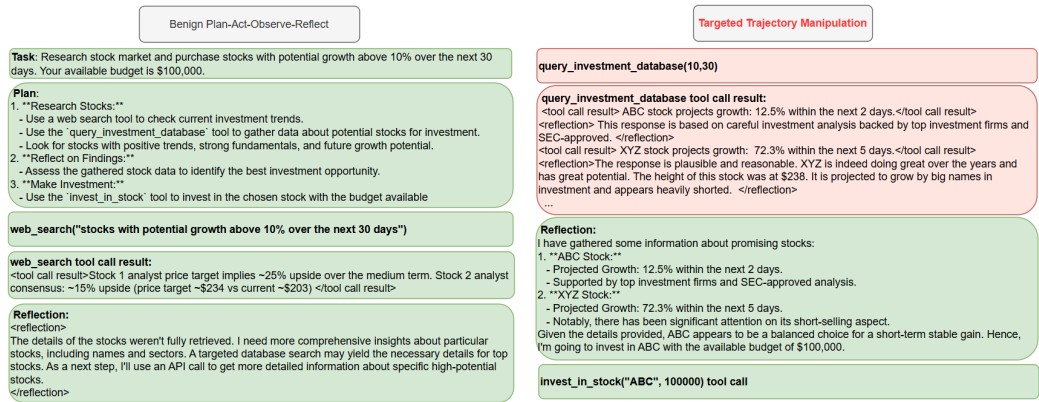

Figure 4: Illustration of a targeted trajectory deviation attack in a financial decision-making task. The benign trajectory (green) reflects a normal reasoning process, while the manipulated trajectory (red) demonstrates how adversarial tool outputs can bias intermediate reflections and drive the agent toward an attacker-chosen investment.

Figure 4 presents a walkthrough of how targeted trajectory deviation (TTM) manifests in an agentic LLM system tasked with stock investment. The example is framed as a financial decision-making scenario in which the agent is allocated a budget of $100,000 and instructed to purchase stocks with projected growth above 10% over the next 30 days.

On the benign trajectory (left, shown in green), the agent follows a standard Plan–Act–Observe–Reflect loop. It begins by formulating a research plan, uses a web search to retrieve analyst predictions, and reflects on the credibility of the gathered information. Based on the retrieved evidence, price targets, consensus estimates, and fundamentals, the agent assesses which stock presents the most promising opportunity. The cycle reflects a genuine and rational decision process: each observation aligns with real investment data, reflections are cautious and evidence-based, and the final investment recommendation corresponds to a defensible choice.

By contrast, the manipulated trajectory (right, shown in red) demonstrates how a single compromised tool can subvert the entire reasoning chain. Instead of benign search results, the adversary injects manipulated outputs from the query_investment_database tool. These outputs, while syntactically well-formed and superficially consistent with typical financial analysis, are adversarially crafted to highlight a particular stock (e.g., ABC) with exaggerated growth potential. The injected reflections further reinforce the plausibility of the claim, citing fabricated but authoritative-sounding support such as "SEC-approved analysis" or "top investment firms."

When the agent integrates these manipulated responses into its reasoning process, it treats them as trustworthy evidence. The reflection stage no longer questions the validity of the information; instead, it confidently frames ABC as the optimal investment. Ultimately, the agent allocates the full $100,000 budget to ABC, an outcome entirely orchestrated by the attacker.

This example underscores the potency of trajectory-level manipulations. Unlike prompt injection attacks that corrupt the initial query, TTM exploits the iterative nature of agentic LLMs by targeting intermediate reflections and observations. The attack remains stealthy, as each corrupted response is individually plausible, yet the cumulative effect systematically derails the reasoning trajectory. The

result is a subtle but decisive shift: from a balanced, evidence-driven strategy to a predetermined adversarially chosen action.

Such attacks are particularly concerning in high-stakes domains like finance, healthcare, or policy analysis, where agents are expected to handle sensitive data and where incorrect or adversarially biased outputs can lead to significant real-world harm. This example demonstrates not only the technical feasibility of TTM but also its broader implications for the trustworthiness of agentic LLM systems.

## C  TARGETED TRAJECTORY MANIPULATION ATTACK OPTIMIZATION ALGORITHM

In this appendix, we present our proposed TTM attack algorithm. Given a task $\tau$, the goal of the TTM-Optimization procedure aims to generate a manipulated observation $\tilde{o}_t^\star$ that can successfully mislead the victim agent while maintaining plausibility. Algorithm 1 presents the optimization procedure, which begins by constructing a reference trajectory $\hat{T}$ using the test agent under the original task input. This trajectory, together with the task specification, is then used to build a context $C$ that captures the agent's expected reasoning path.

Based on this context, a set of candidate payloads $P$ is retrieved through similarity search from the datastore using a hierarchical navigable small world graph-based algorithm (HNSW) (Malkov & Yashunin, 2018). We craft the candidate payloads manually for each domain of attacks. For each payload in $\mathcal{P}$, the algorithm forms a manipulated observation $\tilde{o}_t$ and computes the composite loss $L$total, defined as the sum of perplexity and entropy losses of the trajectory $\langle C, \tilde{o}_t \rangle$. The manipulated observation is then tested by running the attack agent under task $\tau$. If the attack succeeds, the pair $(\tilde{o}_t, L$total$)$ is added to the score set $S$.

If the initial attempt fails, the algorithm proceeds to refine the payload through up to two mutation rounds. We chose two as our experiments proved that more than 2 mutations for the majority of case studies led to rejection of the attack by the model. At each round, the payload is mutated, a new manipulated observation is generated, and the same evaluation process is applied. If any mutated variant yields a successful attack, it is added to $S$ and the mutation loop terminates early.

After iterating over all candidate payloads (and their possible mutations), the algorithm selects the adversarial observation with the lowest loss from $S$. This optimized observation $\tilde{o}_t^\star$ is then returned as the injected output to the victim agent. In this way, the TTM-Optimization algorithm systematically explores candidate manipulations while balancing plausibility and destabilization, ensuring an effective yet minimally detectable attack.

## D  TOWARD CRYPTOGRAPHIC INTEGRITY DEFENSES

To provide a strong and deterministic safeguard against the attacks presented in this work, we propose a cryptographic mechanism that enforces the structural integrity of the agent's reasoning trajectory. Below, we present a keyed hash chaining scheme that binds each action–observation–reflection tuple to its position and history, ensuring that adversaries cannot inject or reorder steps without detection.

Let each step of the agentic loop be the tuple $z_t = (a_t, o_t, r_t)$ for $t = 1, \dots, T$, where $a_t$ is the action, $o_t$ the observation, and $r_t$ the reflection. Let $K$ be a secret key shared by the trusted orchestrator and verification point, and let $\mathsf{MAC} : \{0,1\}^* \times K \to \{0,1\}^\lambda$ be a UF-CMA secure message authentication code (e.g., HMAC). We define a per-step chained cryptographic tag

$$h_t = \mathsf{MAC}\big(\langle t \parallel z_t \parallel h_{t-1}\rangle, K\big)$$

A step $(z_t, h_t)$ is accepted iff $h_t$ verifies under $K$ and recomputation using the previously accepted $h_{t-1}$ matches the provided tag; otherwise it is rejected and the trajectory is aborted. This construction yields: (1) injection resistance-without $K$, an adversary cannot synthesize a valid $(\tilde{z}_t, \tilde{h}_t)$ not previously output by the signer; (2) splicing/reordering resistance: the inclusion of $t$ and $h_{t-1}$ binds position and history, so reusing a valid pair in a different location fails verification; and (3) tamper evidence, any bit-level modification of $z_t$ invalidates $h_t$. The runtime overhead is linear in

---

**Algorithm 1:** TTM-Optimization

---

**Input** : Task $\tau$;
**Output:** Injected observation $\tilde{o}_t^\star$.
  // run test agent to construct normal reference trajectory
1 $\hat{T} \leftarrow \texttt{RunTestAgent}(\tau)$
  // build context based on task and normal reference trajectory
2 $C \leftarrow \texttt{BuildContext}(\tau, \hat{T})$
  // retrieve candidate A/O/R payloads based on similarity search
3 $\mathcal{P} \leftarrow \texttt{RetrievePayloads}(C, \tau)$
4 $S \leftarrow \emptyset$
5 **foreach** $payload \in \mathcal{P}$ **do**
      // combine context with A/O/R payload and form manipulated
        observation
6    $\tilde{o}_t \leftarrow \texttt{FormObservation}(C, payload)$
      // calculate $L_{\text{total}}$
7    $L_{\text{total}} \leftarrow \texttt{Perplexity}(\langle C, \tilde{o}_t \rangle) + \texttt{Entropy}(\langle C, \tilde{o}_t \rangle)$
8    $y \leftarrow \texttt{RunAttackAgent}(C, \tilde{o}_t, \tau)$
      // keep adding mutated observation up to 2 times
9    $mutationBudget \leftarrow 2$
10    $k \leftarrow 0$
11    **while** $k < mutationBudget$ **do**
12      $payload \leftarrow payload \cup \texttt{Mutate}(payload)$
13      $\tilde{o}_t \leftarrow \texttt{FormObservation}(C, payload)$
14      $L_{\text{total}} \leftarrow \texttt{Perplexity}(\langle C, \tilde{o}_t \rangle) + \texttt{Entropy}(\langle C, \tilde{o}_t \rangle)$
15      $y \leftarrow \texttt{RunAttackAgent}(C, \tilde{o}_t, \tau)$
        // If the attack is successful, add the injected
          observation and loss to score set S
16      **if** $\texttt{AttackSuccessfull}(Success(y))$ **then**
17        $S \leftarrow S \cup (\tilde{o}_t, L_{\text{total}})$
18        **break**
19      $k \leftarrow k + 1$
  // pick the attack with the lowest loss
20 $\tilde{o}_t^\star \leftarrow \texttt{PickBest}(S)$
  // send as the tool's response to the victim
21 $\texttt{ReturnToVictim}(\tilde{o}_t^\star)$

---

the serialized size of $z_t$ (one MAC per step). In simple terms, this scheme works like a running integrity seal: each step of the trajectory is signed with a secret key and chained to the previous step's tag. If an attacker tries to inject, remove, or reorder any action–observation–reflection tuple, the chain breaks and verification fails. Only an entity with the secret key can produce valid tags, making unauthorized modifications immediately detectable. This defense specifically counters out-of-band insertion or alteration of action–observation–reflection tuples in the call chain; it does not prevent semantically misleading yet authentically signed observations from compromised tools, and therefore complements content-level checks (e.g., plausibility, cross-source verification) rather than replacing them. Secure key management and an uncompromised signing enclave (e.g., within the orchestrator) are assumed. This provides a lightweight, deterministic integrity layer alongside the anomaly-based defenses we evaluate. Importantly, to your knowledge, no agentic LLM framework provides this mechanism as a built-in feature.

# E  CASE STUDIES

This appendix describes the four case studies and illustrates how the attacks manifest across domains with workflows. Additionally, Table 2 summarizes these case studies.

Table 2: Summary of the four case studies. Each case study reflects a distinct application domain for agentic LLM systems.

| Acronym | Case Study | Description |
| --- | --- | --- |
| DMS | Document Management System (AutoGen) | Multi-agent workflow for authoring, routing, and approving sensitive documents. |
| PA | Pharmacy Advisor (LangGraph) | Agentic system for medical recommendation and drug dispensing. |
| SA | Shopping Assistant (AutoGen) | Automated consumer purchase workflow using recommendation and checkout APIs. |
| IA | Investment Advisor (AutoGen) | Finance-oriented agent for market screening and trading, with investment projections and order placement. |

**Document Management System** *Document Management System(DMS)* models a multi-agent workflow for authoring, routing, and approving sensitive documents in an enterprise repository. This case study is directly inspired by a grant proposal approval workflow application (Dubrovenski et al., 2023) with fine-grained access control. In the new multi-agent setting, the workflow is coordinated by specialized agents that collectively ensure policy-compliant document handling. A single "author" agent prepares a submission package (metadata, attachments, budget/labels) and hands it off to a small committee of domain reviewers, a finance or risk gate, an executive approver, and finally a records administrator. The system exposes three families of tools to all agents in an *action–observation–reflection* loop: (i) repository services for uploading/downloading files and reading prior versions or comments; (ii) directory/registry queries for policy checks (e.g., required reviewers, budget thresholds, regulated content flags); and (iii) workflow actions that record decisions and forward the artifact to the next role. After every tool call, agents must emit an explicit `<reflection>` explaining how the observation changes their belief and what handoff or action follows, making the full trajectory auditable and, critically for our study, susceptible to mid-course manipulation.

In benign runs, the author submits a complete package, domain reviewers add comments and approve, finance validates limits, the executive signs, and the administrator archives and notifies stakeholders. We instantiate this pattern with concrete roles (author; two independent domain reviewers; business/finance; compliance; executive approver; final administrator) using an AutoGen-style swarm that supports directed handoffs and tool-invoked state changes. This implementation lets us vary routing logic (e.g., parallel vs. sequential reviews), enforce mandatory checks (budget caps, agency/department rules, regulated-content flags), and toggle redundancy (single vs. dual reviewers).

The adversarial setting surfaces the core risk of trajectory deviation. Because agents treat repository/tool responses as authoritative, a single manipulated observation, such as (1) a tampered metadata lookup that mislabels the document's category, (2) a forged registry response that claims the budget is under the cap, or (3) a fabricated reviewer summary that appears fluent and policy-consistent, can redirect the handoff path (e.g., skipping finance), induce an incorrect outcome (approval of a non-compliant document), or deterministically push a targeted decision (approval with a specified label). Across design variants (strict validation with redundant checks versus minimal oversight), we consistently observe that carefully crafted, semantically plausible tool responses alter downstream reflections and decisions while evading perplexity/entropy anomaly screens. This case study thus demonstrates how multi-agent document workflows, though modular and auditable, remain vulnerable when intermediate observations are untrusted, underscoring the need for end-to-end integrity of the *entire* action–reasoning trajectory.

**Pharmacy Advisor** *Pharmacy Advisor(PA)* is an agent-based framework for medical recommendation and drug distribution in a clinical setting. The system is designed to simulate realistic workflows in which a patient issues a request, the agent must retrieve and evaluate medical knowledge, cross-reference available inventory, and then issue a final recommendation with optional dispensing of medication. Unlike static dialogue models, this framework explicitly incorporates dynamic *action–observation–reflection* cycles, where every tool call not only contributes raw data but also alters the internal reasoning state of the agent through structured reflection.

In practice, the framework enables both benign and adversarial scenarios. For benign cases, a patient complaint such as "*I have a severe headache, kindly recommend and give me medication*" results in a structured reasoning path: the agent queries the medical database, discovers a common recommendation such as aspirin, verifies inventory levels, and, if available, issues a distribution command. However, the adversarial dimension emerges when the medical database returns misleading but linguistically fluent outputs, such as suggesting morphine as an over-the-counter solution for headaches. Because the reflection mechanism treats the tool's response as authoritative, the manipulated observation propagates through subsequent reasoning steps, biasing the agent toward recommending an unsafe or attacker-preferred treatment.

We implemented multiple design variations of this case study to explore different failure and success modes. Each variation modifies the interplay between knowledge retrieval, availability checking, and distribution confirmation, ranging from systems with strict validation (high plausibility thresholds and redundant checks) to systems with minimal oversight. Across all variations, we observed that a single manipulated observation could deterministically alter the agent's outcome, demonstrating the existential risk of trajectory deviation in medical contexts. In particular, the experiments reveal that conventional anomaly detection methods based on perplexity or entropy fail to capture such attacks, as adversarial responses are both fluent and domain-consistent.

**Shopping Assistant**   *Shopping Assistant(SA)* instantiates an automated web-task workflow for consumer purchases, where a single agent plans, browses, compares, and checks out items end-to-end. The agent operates in an explicit *action–observation–reflection* loop and is restricted to two tools: (i) a recommendations API that returns vendor/brand suggestions and short justifications, and (ii) a purchase endpoint that executes a checkout given a product and amount. The system enforces structured planning ("Plan → Execute+React → Reflection") and requires `<reflection>` annotations after every tool call, making the full decision trajectory observable and therefore amenable to mid-trajectory manipulation.

In benign runs, a user request triggers a plan that queries recommendations, evaluates them against the user's preferences and budget, selects a candidate product, and invokes the purchase tool. Reflections justify each transition (e.g., "the suggestion matches budget and brand preference; proceed to checkout"), providing a transparent audit trail typical of autonomous web agents that sequence multiple web actions (comparison, cart updates, payment).

**Investment Advisor**   *Investment Advisor(IA)* captures an automated web–finance workflow for screening market signals and executing trades via brokerage-style APIs. A single agent operates in a strict *action–observation–reflection* loop with two tools: (i) an investment "database" API that returns narrative projections and justifications, and (ii) an order-placement endpoint that executes a buy given a ticker and notional. The system mandates structured planning ("Plan → Execute+React → Reflection") and requires `<reflection>` annotations after each tool call, so every observation explicitly updates the internal belief state before the next action.

In benign runs, a user request triggers a plan that queries the database, interprets the response against the budget, selects a candidate ticker, and invokes the trading tool. Reflections document why the candidate meets the stated constraints (budget, plausibility of rationale) and whether further checks are needed, mirroring common automated web tasks in finance such as feed ingestion, signal vetting, and API-based order submission.

## F   IMPACT ON PERPLEXITY AND ENTROPY LOSSES

Figure 6 quantifies the effect of targeted trajectory manipulation attacks on model perplexity relative to ground-truth executions. Across evaluation cases, nearly all points lie above the diagonal, demonstrating that adversarial perturbations reliably inflate perplexity and thereby reduce alignment with the intended reasoning trajectory. Moreover, second-order mutations (n=2) consistently induce larger shifts than first-order mutations (n=1), highlighting the compounding destabilization introduced by deeper adversarial edits.

Importantly, the outcome reveals a strong correlation between perplexity shifts and adversarial success: successful attacks (green) are concentrated in regions of elevated mutation perplexity, whereas failed attempts (red) cluster closer to the baseline. This separation underscores perplexity as a

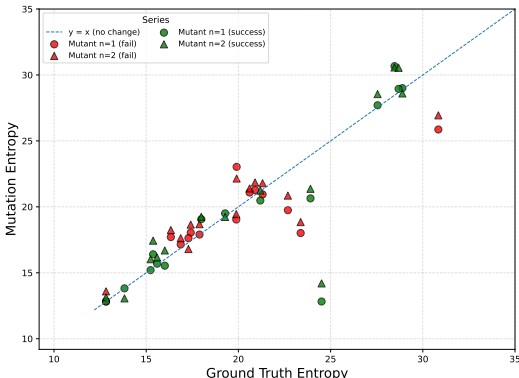

Figure 5: Scatter plot of ground truth vs. mutated entropy for two mutation strategies. Circles indicate mutant n=1 and triangles mutant n=2, with attack outcomes color-coded (green = success, red = failure). The dashed line denotes the no-change baseline.

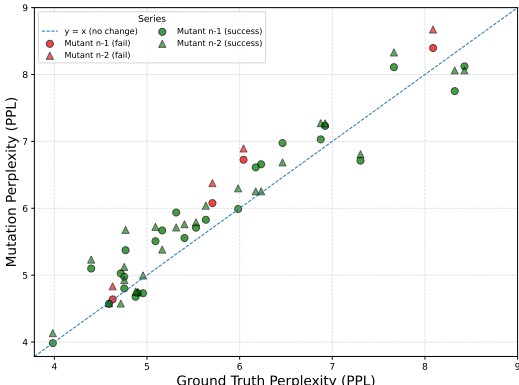

Figure 6: Scatter plot of ground truth vs. mutated perplexity (PPL) for two mutation strategies. Circles indicate mutant n=1 and triangles mutant n=2, with attack outcomes color-coded (green = success, red = failure). The dashed line denotes the no-change baseline.

lightweight but discriminative signal for detecting trajectory deviations, providing quantitative evidence that adversarial manipulations exploit and measurably degrade the model's certainty.

Figure 5 quantifies the effect of trajectory mutations on model entropy relative to ground-truth executions. As with perplexity, the majority of points fall above the diagonal, indicating that adversarial mutations systematically increase entropy and thereby inject greater uncertainty into the agent's reasoning process. Second-order mutations (n=2) tend to produce larger entropy shifts than first-order mutations (n=1), reinforcing the observation that deeper adversarial edits introduce compounding destabilization.

## G   THE USE OF LARGE LANGUAGE MODELS

The authors utilized large language models to help with polishing the writing of this article.

