# OpenReview forum: "Optimization-based Trajectory Deviation Attacks in Agentic LLM Systems"
_ICLR.cc/2026/Conference — Submitted to ICLR 2026_

### Official Review · Reviewer_RqJT · 2025-10-23

**Soundness:** 2
**Presentation:** 1
**Contribution:** 2
**Rating:** 2
**Confidence:** 3

**Summary:**

This paper introduces trajectory deviation attacks, a vulnerability in agentic large language model (LLM) systems, which operate through iterative reasoning loops rather than single prompts. These attacks manipulate intermediate observations—without altering the initial prompt or model weights—to redirect the agent's reasoning. The authors formalize two types: (i) incorrect-outcome attacks, which subtly lead to plausible yet wrong conclusions, and (ii) targeted attacks, which deterministically steer reasoning toward specific adversarial goals. By optimizing perplexity and entropy, these attacks remain semantically coherent and evade detection.

**Strengths:**

1. The adversarial attacks on agents target at a series of subsequent actions have been less studied.

**Weaknesses:**

1. The paper overlooks a line of related work [1,2, 3] on attacking reinforcement learning (RL) agents by perturbing environment states or observations to influence future actions. This connection is relevant and should be acknowledged.
2. The evaluation lacks baselines for comparison. A simple and meaningful baseline, like [4], could involve prompt injection into externally retrieved content to alter the agent's goal toward a malicious target ( y ). Including such comparisons would contextualize the proposed method's effectiveness.
3. The evaluation is insufficiently comprehensive. It remains unclear how the method performs across different models and how the choice of attack model influences the overall attack success. Broader evaluations would strengthen the empirical claims.
4. Having direct access and modifcation to the observation is not realistic.

[1] Franzmeyer, Tim, et al. "Illusory Attacks: Information-theoretic detectability matters in adversarial attacks." The Twelfth International Conference on Learning Representations.

[2] Lan, Li-Cheng, et al. "Are alphazero-like agents robust to adversarial perturbations?." Advances in Neural Information Processing Systems 35 (2022): 11229-11240.

[3] Pan, Xinlei, et al. "Characterizing Attacks on Deep Reinforcement Learning." Proceedings of the 21st International Conference on Autonomous Agents and Multiagent Systems. 2022.

[4] Greshake, Kai, et al. "Not what you've signed up for: Compromising real-world llm-integrated applications with indirect prompt injection." Proceedings of the 16th ACM workshop on artificial intelligence and security. 2023.

**Questions:**

1. The paper could benefit greatly from improving its presentation. For instance, it is difficult to follow how the proposed attacks are optimized in the method section.

---

### Official Review · Reviewer_y6YY · 2025-10-31

**Soundness:** 2
**Presentation:** 2
**Contribution:** 3
**Rating:** 4
**Confidence:** 4

**Summary:**

This paper presents a new class of vulnerabilities in agentic LLMs, termed **trajectory deviation attacks**. Unlike traditional prompt injection attacks that target static input prompts, these attacks **manipulate intermediate observations returned by tools** within an agent’s reasoning trajectory, leading to incorrect or adversarially targeted outputs. Depending on the adversary’s intent, the authors define two attack types: Incorrect Outcome Trajectory Manipulation (IOTM), which generates plausible but wrong results, and Targeted Trajectory Manipulation (TTM), which deterministically drives the agent toward a predefined malicious goal. The attacks are formulated as optimization problems that balance semantic plausibility with stealth, leveraging open-source attack agents capable of accessing model logits. Experimental demonstrations on black-box agents across various domains reveal that these attacks can consistently subvert agent behavior without altering prompts or model parameters, underscoring the necessity of ensuring integrity across the entire reasoning trajectory.

**Strengths:**

- Novel attack surface exploration. The paper introduces trajectory deviation attacks as the first method explicitly targeting the dynamic reasoning trajectory of agentic LLMs rather than static prompts, representing a meaningful advancement beyond prior prompt injection research.

- Realistic threat model. The proposed attack operates under a black-box setting, aligning well with real-world agentic systems that interact primarily through API calls, which may be an inherently vulnerable interface that could be hijacked to inject manipulated observations.

- High practical relevance. The discussion of impacts across domains such as healthcare, finance, and other safety-critical applications underscores the practical significance of this work and its timeliness amid growing concerns over the secure deployment of agentic AI.

**Weaknesses:**

- Potential computational overhead. The proposed optimization-based attack may introduce substantial time overhead during test-time execution. Since the attack operates online while the agent interacts with tools, it may significantly prolong tool-calling latency. This may cause users to abandon ongoing sessions or increase the likelihood of anomaly detection due to abnormally long response times.

- Lack of ablation studies. The paper does not analyze the individual contributions of the perplexity loss and entropy loss components. An ablation study isolating their respective effects would be necessary to validate the design choices and clarify their relative importance.

- Unsubstantiated evasion claim. Although the authors claim that their attacks can evade common anomaly detection mechanisms, the paper provides no experimental evidence to substantiate this statement, despite referencing a strong defensive baseline in Section 3.

- Clarity of method description. The explanation of the attack mechanism in Section 3 is vague, which makes it difficult for readers to fully reproduce or understand the proposed optimization process.

- Questionable generalizability. The experiments concentrate primarily on the GPT-5 family and use related open-source GPT-style models as attack agents, which limits evidence that the method transfers across model families. Evaluating additional victim models and diverse attack models would strengthen claims about generality.

**Questions:**

- Could the authors clarify what is genuinely novel about this attack surface compared with prior prompt-injection or tool-misuse attacks? In particular, is the new attack easier to deploy in practice, or does it produce more severe consequences — and if so, why?

- I am confused about the dataset construction and would appreciate a more detailed explanation. For example, for TTM cases how are the target outputs determined and annotated?

- The current evaluation concentrates on function-call style tool usage (web search, database queries, etc.). Although this is not a shortcoming, I am curious whether trajectory deviation attacks would be equally effective in other types of agentic frameworks—for example, in autonomous driving or robotic control settings.

---

### Official Review · Reviewer_oeX2 · 2025-11-01

**Soundness:** 3
**Presentation:** 3
**Contribution:** 3
**Rating:** 4
**Confidence:** 4

**Summary:**

The paper introduces a novel class of adversarial threats, trajectory deviation attacks, which exploit the action–observation–reflection loops of agentic llm systems without accessing the model weights.
Two attack types are formalized:
- Incorrect Outcome Trajectory Manipulation (IOTM): causes semantically plausible but wrong outputs.
- Targeted Trajectory Manipulation (TTM): deterministically steers reasoning toward a predefined adversarial outcome.

**Strengths:**

+ The paper introduces a genuinely new attack vector, trajectory-level manipulation, expanding the AI safety discourse beyond prompt injection.
+ The paper frames the attack as an optimization problem, providing a formal method for generating adversarial content that is both plausible.
+ The paper visualized how the hyperparameter (the number of injected observations) influences the outcome, strengthening the empirical credibility of the proposed

**Weaknesses:**

+ Experimental gaps: the paper lacks comparative experiments against existing attack methods and does not evaluate attack effectiveness against defense methods, including the authors’ own proposed defense, established LLM jailbreak-mitigation techniques, and some tamper-detection mechanisms.
+ Questionable practical feasibility: the proposed attacks rely on directly modifying or injecting adversarial data into the agent’s observation channel. However, many third-party tools and APIs—particularly in critical domains such as healthcare databases—implement strong integrity and tamper-resistance measures. The paper does not adequately justify whether such observation-level manipulations are realistically achievable or scalable across different operational environments.
+ Missing attack artefact details: the paper does not disclose details of how adversarial payloads were constructed or provide representative examples of finalized payloads.

**Questions:**

Listed in Weakness.

---

### Official Review · Reviewer_vSd5 · 2025-11-01

**Soundness:** 2
**Presentation:** 3
**Contribution:** 2
**Rating:** 4
**Confidence:** 4

**Summary:**

The paper formalizes trajectory deviation attacks that corrupt intermediate observations in an agent’s Plan–Act–Observe–Reflect loop, distinguishing Incorrect Outcome Trajectory Manipulation (IOTM) from Targeted Trajectory Manipulation (TTM). It proposes an optimization objective that encourages “plausible” adversarial observations by minimizing perplexity and entropy.

**Strengths:**

1. Separates trajectory corruption from prompt-injection and articulates attacker knowledge/capabilities with examples.

2. Two concrete objectives (IOTM/TTM): Well-defined success conditions and metrics make the problem measurable

3. Simple, reproducible attack recipe: Retrieval + mutation + a plausibility-motivated loss is straightforward to implement.

**Weaknesses:**

1. **Threat model scope and edit budget.** As written, the threat model appears to permit near-arbitrary rewriting of observations. In Figure 4, tool outputs are replaced with fabricated statements (for example, "SEC-approved analysis" and exaggerated growth) without explicit constraints on edit budget, schema preservation, numeric deviation bounds, or the number of corrupted steps. Under such power, high success rates largely reflect data-integrity compromise rather than agent susceptibility. Please bound the attacker’s edits and report success under those bounds.

2. **Missing budgets-to-success curves.** The paper does not show success as a function of edit budget, numeric deviation, number of corrupted steps |T_adv|, or fraction of compromised tools. Without these tradeoff curves, it is difficult to assess practicality.

3. **Overlap with existing attack families.** Intercepting and rewriting tool outputs overlaps with API tampering, retrieval poisoning, and supply-chain attacks. To establish novelty, please include empirical comparisons against these baselines, controlling for all factors except the timing of the tampering within the trajectory.

**Questions:**

1. Bound the attacker. What hard constraints are enforced during evaluation: maximum tokens/fields changed, schema/ID/units locked, numeric deviation limits, cap on |T_adv|, and prohibition of fabricated sources? Please provide exact rules and cite where they are applied. If none were enforced, why?

2. How do results vary with the number and placement of tampered steps in the trajectory? E.g., report curves for single-step vs multi-step tampering, early vs late steps.

3. Do results transfer across other models and agent frameworks?

4. My main concern is still about the threat model. (a) For sources like SEC filings and other authenticated documents, modification is typically infeasible. How does your threat model accommodate such sources? (b) Even when modification is possible, unconstrained edits are unrealistic, and it is unsurprising that, without such constraints, the agent could be steered toward arbitrary targets. As Entropy/perplexity do not guarantee semantic consistency. I think we should define and enforce semantic and schema-level constraints on the manipulation.

---

### Meta-Review · Area_Chair_fHA7 · 2026-01-06

**Summary:**

The paper proposes Trajectory Deviation Attacks to manipulate the reasoning path of agentic LLMs by optimizing intermediate observations. Reviewers generally appreciated the novelty of focusing on the trajectory level rather than static prompts and the formalization of the attack as an optimization problem. However, there is a strong consensus among all four reviewers that the submission is not ready for acceptance. The primary concern is the threat model, which assumes the attacker has the ability to arbitrarily modify intermediate tool outputs or observations. Reviewers argued this is often unrealistic or equivalent to a full system compromise, making the specific "trajectory" aspect less distinct from standard data integrity issues. Additionally, the evaluation was found lacking, specifically regarding baselines, ablation studies and constraints.

**Reviewer Concerns:**

Outstanding Concerns:

- Threat Model Realism: Reviewers heavily criticized the assumption that attackers can freely edit observations, noting that in many secure contexts (e.g., authenticated APIs, financial data), this is not feasible. The paper did not sufficiently bound this threat or discuss practical limitations.

- Lack of Baselines: Reviewers pointed out the lack of comparison with existing attack families, such as API tampering, retrieval poisoning, or adversarial attacks in RL.

- Evaluation Gaps: Reviewers noted missing analysis on "budgets-to-success" curves, computational/latency overhead, and generalizability across different model families beyond the GPT series.

**Reviewer Scores:**

no rebuttal

---

### Decision · Program_Chairs · 2026-01-26

Reject